# Metabolomics Approach to Characterize Green Olive Leaf Extracts Classified Based on Variety and Season

**DOI:** 10.3390/plants11233321

**Published:** 2022-12-01

**Authors:** Graziana Difonzo, Maria Assunta Crescenzi, Sonia Piacente, Giuseppe Altamura, Francesco Caponio, Paola Montoro

**Affiliations:** 1Department of Soil, Plant and Food Science, University of Bari Aldo Moro, Via Amendola, 165/a, I-70126 Bari, Italy; 2Dipartimento di Farmacia, Università Degli Studi di Salerno, Via Giovanni Paolo II, 132, I-84084 Fisciano, Italy; 3PhD Program in Drug Discovery & Development, Pharmacy Department, University of the Study of Salerno, I-84135 Salerno, Italy; 4Centro di Ricerca, Sperimentazione e Formazione in Agricoltura Basile Caramia, Locorotondo, I-70010 Bari, Italy

**Keywords:** olive leaf extract, metabolomics, multivariate analysis, polyphenols, oxidation, upcycling

## Abstract

The huge interest in the health-related properties of plant polyphenols to be applied in food and health-related sectors has brought about the development of sensitive analytical methods for metabolomic characterization. Olive leaves constitute a valuable waste rich in polyphenols with functional properties. A (*HR*)*LC-ESI-ORBITRAP-MS* analysis with a multivariate statistical analysis approach using PCA and/or PLS-DA projection methods were applied to identify polyphenols in olive leaf extracts of five varieties from the Apulia region (Italy) in two different seasonal times. A total of 26 metabolites were identified, further finding that although metabolites are common among the different cultivars, they differ in the relative intensity of each peak and within each cultivar in the two seasonal periods taken into consideration. The results of the total phenol contents showed the highest content in November for Bambina and Cima di Mola varieties (1816 and 1788 mg/100 g, respectively), followed by Coratina, Leccino, and Cima di Melfi; a similar trend was found for the antioxidant activity and RapidOxy evaluations by reaching in Bambina values of 45 mmol TE/100 g and 85 min of induction time.

## 1. Introduction

In the Mediterranean area, olive (*Olea europaea* L. subsp. *europaea*) is considered to be one of the oldest and important agricultural crops and is characterized by a large number of cultivars used for the production of olive oil and table olives. In this scenario, Italy and its very rich olive germplasm—estimated to include about 800 cultivars—play a dominant role, not only in the preservation of olive biodiversity but also in the production of high-quality olive oils with strong sensory specificity [1]. The olive oil industry generates a large amount of waste and by-products, including olive pomace, olive mill wastewater, and olive leaves [2]. Olive leaves are always used as animal feed, but they have the potential to be used in other applications with higher added value such as in cosmetic, therapeutic, and food industries [3]. In recent years, the incorporation of leaves during the extraction of olive oils was evaluated, and it was found that the usage of leaves can enhance the chemical–sensory quality of olive oils [4]. Olive leaves have long been known for their therapeutic and medicinal properties and they are used in both traditional and modern medicine; they are an important source of phenolic compounds, such as oleuropein, verbascoside, hydroxytyrosol, and tyrosol, which all exhibit important biological activity—for example, antioxidant and antimicrobial properties [5,6]. Thus, the exploitation of olive leaves as a source of polyphenols might bring additional benefits to the olive sector by providing an additional source of income for producers and adding value to the supply chain.

Olive leaves are available on the trees throughout the year but are available in larger quantities during late winter and early spring as pruning residues or as by-products separated from fruits before processing in autumn. Recent studies revealed a significant impact of geographical origin and showed that polyphenols in olive leaves of particular cultivars are more sensible to pedoclimatic variations than others [7,8]. In fact, seasonal variation in chemical compositions is a well-known phenomenon in plants, and it is associated with the biosynthesis, stability, and degradation of secondary metabolites in olives. In addition, quantitative and qualitative changes in the biochemical composition of olive leaves also depends on the plant variety, climatic conditions, sampling time, genetics, and geographical origin [9,10]. Therefore, it would be of practical importance to investigate the variation in polyphenols in the extracts produced from olive leaves collected from different Apulian cultivars both from pruning and olive harvesting in order to upcycle this waste.

Recent advances in multi-omics approaches have enabled mining and mapping of a large number of datasets at different biological scales of living organisms.

Liquid chromatography coupled with high-resolution tandem mass spectrometry (LC-(HR)MS^n^) is able to separate, fragment, and characterize most of the plant metabolites from a vegetable source [11,12]. This technique gives advanced information when applied and coupled with multivariate statistics and pathway analysis to develop a metabolomics approach [13,14,15]. Thus, the chromatographic and mass spectrometric resulting data can be processed with a multivariate statistical analysis approach, using PCA and/or PLS-DA projection methods. This approach allows the classification of different samples in relation to metabolites, characterizing them as markers.

In this work, we characterized the metabolite profiles of olive leaf extracts from five cultivars grown in the Apulia region (southern Italy) in two different seasons by *LC-ESI*/*LTQ ORBITRAP*/*MS* and *LC-ESI*/*LTQ ORBITRAP*/*MS*/*MS* coupled with multivariate data analyses. Furthermore, the main detected compounds were quantified and the extracts were tested for antioxidant activity and oxidative stability.

## 2. Results and Discussion

### 2.1. LC-ESI/LTQ-Orbitrap/MS and LC-ESI/LTQ-Orbitrap/MS/MS Analysis

To investigate the main secondary metabolites and their profiles in the different olive leaf extracts, an LC method coupled with a hybrid mass spectrometer, combining linear trap quadrupole (LTQ) and an Orbitrap mass analyser, was developed and used in this study. Negative ion mode was selected for the generation of spectra due to its better sensitivity for most of the phenolic compounds investigated.

In order to obtain a full identification for all metabolites, LC-ESI-LTQ-Orbitrap/MS/MS experiments were run by using data-dependant analysis (DDA) by selecting high-resolution experiments for precursor ions and low-resolution experiments for MS/MS spectra results. Chromatograms obtained from the hydro-alcoholic extracts recovered from the olive leaves collected in April and the olive leaves collected in November are reported in Figure 1. The profiles showed small variations in relation to the collection time, and the comparison between the chromatograms obtained analysing extracts from different leaves showed consistent differences among the cultivars, prompting investigations to be carried out on the metabolite responsible for the observed specificity. The peaks are numbered based on the identification of the peaks that are reported in Table 1.

The results obtained from the compound identification by LC-ESI-Orbitrap/MS and LC-ESI-Orbitrap/MS/MS are reported in Table 1. The phenolic compounds were identified on the basis of retention times, accurate mass measurements, and fragmentation and by exploring public mass spectral data repositories of specific metabolites (Mass Bank) and using literature data as a comparison [16,17,18].

**Table 1 plants-11-03321-t001:** Metabolites identified in olive leaves extracts by LC-ESI/LTQ-Orbitrap/MS and LC-ESI/LTQ-Orbitrap/MS/MS analysis.

N°	RT	[M-H]^−^	Molecular Formula	Δppm	MS/MS	Identity	CO	BA	MO	ME	LE	References
1	1.74	341.1083	C_12_H_21_O_11_	1.3	179.01	sucrose	X	X			X	[19]
2	3.13	191.0193	C_6_H_7_O_7_	3.7	111.13	citric acid	X	X	X	X	X	[20]
3	8.35	375.1285	C_16_H_23_O_10_	−0.2	330.99	loganic acid	X	X	X		X	[18]
4	10.85	315.1077	C_14_H_19_O_8_	0.7	153.06	hydroxytyrosol glucoside	X	X	X	X	X	[18]
5	11.33	315.1081	C_14_H_19_O_8_	1.4	153.06	hydroxytyrosol glucoside isomer I	X	X	X	X	X	[18]
6	11.97	389.1073	C_16_H_21_O_11_	−1.5	139.15/165.13/208.95	secologanoside	X	X			X	[18]
7	19.07	389.1074	C_16_H_21_O_11_	−1.2	165.27/181.09/209.04	oleoside	X	X	X	X	X	[18]
8	21.26	403.1229	C_17_H_23_O_11_	−1.2	371.16/222.93/179.14	elenolic acid glucoside	X	X	X	X	X	[18]
9	23.42	593.1489	C_27_H_29_O_15_	−2.0	353.13/473.01/503.14	vicenin II		X	X	X	X	[19]
10	25.31	403.1235	C_17_H_23_O_11_	0.1	371.16/222.93/179.14	elenolic acid glucoside isomer I	X	X	X	X	X	[18]
11	25.36	415.1598	C_19_H_27_O_10_	−0.1	130.97/149.09/190.95	phenethyl beta-primeveroside	X	X	X	X	X	[21]
12	25.74	461.1646	C_20_H_29_O_12_	−1.7	315.20/297.14/135.16	decaffeoyl verbascoside	X			X	X	[21]
13	30.44	609.1448	C_27_H_29_O_16_	−0.3	301.07	rutin	X	X	X	X	X	[22]
14	31.30	623.1968	C_29_H_35_O_15_	−0.4	461.15	verbascoside	X	X	X	X	X	[21]
15	31.72	447.0916	C_21_H_19_O_11_	−1.4	301.11	quercetin rhamnoside	X				X	[21]
16	32.54	543.2073	C_25_H_35_O_13_	0.1	513.0994/525.0561	dihydrooleuropein	X	X	X	X	X	[18]
17	33.18	701.2278	C_31_H_41_O_18_	−1.3	539.2022	oleuropein diglucoside	X	X	X	X	X	[23]
18	33.82	701.2281	C_31_H_41_O_18_	−0.8	539.2022	oleuropein diglucoside isomer I	X	X	X	X	X	[20]
19	35.67	431.0971	C_21_H_19_O_10_	−0.5	269.13	apigenin glucoside	X	X		X	X	[24]
20	36.06	461.1074	C_22_H_21_O_11_	−1.0	299.08/446.01	diosmetin glucoside	X				X	[24]
21	36.41	447.0922	C_21_H_19_O_11_	0.1	285.05	luteolin glucoside	X	X	X	X	X	[24]
22	39.05	539.1761	C_25_H_31_O_13_	1.0	275.05/307.01/376.89	oleuropein	X	X	X	X	X	[24]
23	41.13	539.1757	C_25_H_31_O_13_	−0.3	275.05/307.01/376.89	oleuropein isomer I	X	X	X	X	X	[25]
24	42.97	539.1759	C_25_H_31_O_13_	−0.1	275.05/307.01/376.89	oleuropein isomer II	X	X	X	X	X	[18]
25	44.57	557.2224	C_26_H_37_O_13_	−0.9	185.20/227.08/370.99	dimethyl hydroxy octenoyloxy secologanoside	X		X		X	[18]
26	48.30	523.1808	C_25_H_31_O_12_	−0.3	259.07/291.13/360.93	ligustroside	X	X	X		X	[18]

Abbreviations: CO (Coratina), BA (Bambina), MO (Cima di Mola), ME (Cima di Melfi), and LE (Leccino).

Metabolites detected in the different samples are summarized in Table 1, where compounds are identified according to their increasing retention time (R_T_). Twenty-six metabolites were identified in total in the olive leaf hydro-alcoholic extracts from different cultivars. Identification was possible by using data collected by LC-ESI/LTQ-Orbitrap/MS and LC-ESI/LTQ-Orbitrap/MS/MS experiments.

From the analysis of the different metabolic profiles of the olive leaf extracts, it emerged that although metabolites are common among the different tested cultivars, they differ in the relative intensity of each peak and within each cultivar in the two seasonal periods taken into consideration. In particular, in the leaves collected in April, the oleoside (**7**) and oleuropein diglucoside metabolites together with aglycone (**17**, **18**, **22**) and their isomers were identified in all cultivars, whereas secoiridoid secologanoside (**6**) was found only in the cultivars Leccino, Coratina, and Bambina. Derivatives **4** and **8** (hydroxytyrosol glucoside and elenolic acid) had a ubiquitous distribution, as they are metabolites deriving from the hydrolysis of oleuropein. Similarly, the phenylpropanoid verbascoside (**12**, **14**) and the oleuropein derivative **16** were present in all cultivars.

As for the flavones, **21** (luteolin glucoside) was ubiquitous, **19** (apigenin glucoside) was found in all cultivars except for Cima di Mola, and **20** (diosmetin glucoside) was found only in Coratina and Leccino cultivars.

The metabolic profiles recovered from the leaf samples collected in April compared with those of November show a similarity between the cultivars Leccino, Cima di Mola, and Bambina, while Coratina and Cima di Melfi appear to be different from the others and similar to each other. By comparing metabolic profiles of the leaves collected in November with those collected in April in a semi-quantitative approach, it emerges that the metabolites **7**, **4**, and **8** (oleoside, hydroxytyrosol glucoside, and elenolic acid glucoside) are expressed in major amounts in November leaves, while compounds **14**, **17**, and **26** (verbascoside, oleuropein diglucoside, and ligustroside) are major compounds in the April leaves. These compounds are interconnected by the same metabolic pathway. In fact, the biosynthesis of oleuropein in Oleaceae proceeds via a branching in the mevalonic acid pathway from the secondary metabolism, resulting in the formation of oleosides [26], from which secoiridoids are derived [27]. The prevalent presence reserved for each metabolite in the secoiridoids pathway in olive leaves in the spring or autumn period is discussed previously in literature with, often, controversial attribution [28].

### 2.2. Multivariate Data Analysis

The registered LC/ESI/LTQ-Orbitrap/MS data were then submitted to multivariate data analysis. In order to discriminate the different samples, initially an untargeted multivariate analysis approach was developed. The LC-ESI/LTQ-Orbitrap/MS profiles of the November and April olive leaf extracts were pre-treated using the open-source software mzMine (http://mzmine.sourceforge.net accessed on 21 June 2021). The software creates a data matrix with the samples in the rows and the areas of integrated and normalized peaks in the columns, compensating for the variations in the retention time (Rt) and *m*/*z* values. The data matrix was then subjected to multivariate analysis with SIMCA(+) software using principal component analysis (PCA) as a projection technique: differences between the various types of samples were highlighted by this approach. The score scatter plot showing the spatial distribution of observations (samples) for PCA analysis of April leaves is shown in Figure 2A. The first component expresses 37% of variance and the second 13% of variance. The plot showed a good discrimination between the cultivars: Coratina and Cima di Melfi varieties, located on the left side of the plot, generate two well-differentiated clusters based on their metabolic expression. The lower right quadrant is occupied by the sample extracts from Cima di Mola leaves. On the other hand, the two varieties Bambina and Leccino show a similar metabolic expression, positioning themselves in the upper right quadrant of the plot, even if Leccino samples appear to have similarities with the other varieties, since they are located in the central part of the plot. A loading scatter plot is represented below in Figure 2A and shows which are the metabolites expressed in the April leaves of the different olive cultivars that cause their clustering.

In particular, in the upper left quadrant of the plot, it is possible to observe the main metabolites that characterize the Coratina variety: dihydrooleuropein (**16**) and ligustroside (**26**).

In the lower left quadrant, conversely, it is possible to observe luteolin 7-*o*-glucoside (**21**) as the main metabolite responsible for the clustering of the Cima di Melfi variety.

As regards the right part of the plot, in the lower quadrant it is possible to observe the two main metabolites that characterize the Cima di Mola variety: oleoside (**7**) and oleuropein diglucoside (**17**).

Bambina and Leccino varieties gave a partial overlap, because the metabolic profiles investigated are likely very similar to each other. However, it was possible to identify the presence of elenolic acid glucoside (**8**) as the main metabolite of the Bambina variety, while loganic acid (**3**) as a metabolite of the Leccino variety.

In Figure 2B the score scatter plot obtained from PCA analysis of November leaves is reported.

The first component expresses 46% of the variance and the second one 23% of the variance. Again, there is good discrimination between the various cultivars, with a good clustering of the samples pertaining to the same variety, even if it differences were shown with respect to the April leaves. In the upper part of the plot, by observing from left to right there are the clusters relating to the Coratina, Leccino, and Cima di Mola cultivars. Leccino is also present in the lower right quadrant together with Bambina and Cima di Melfi varieties, but it is positioned in the lower left quadrant.

From the loading plot, represented below in Figure 2B, metabolites (variables) involved in the clustering can be seen.

In particular, in the left upper quadrant it was possible to identify oleoside (**7**), hydroxytyrosol glucoside (**4**), and decaffeoyl verbascoside (**12**) as marker metabolites of the Coratina leaf variety.

In the central area of the upper portion of the plot, elenolic acid glucoside (**8**) and apigenin glucoside (**15**) are identified as metabolites responsible for clustering the samples of the Leccino leaves.

In the right portion of the plot, it is possible to recognize metabolites that characterize the olive leaves of the Cima di Mola variety: phenethyl primaveroside (**11**), oleuropein diglucoside (**17**), and loganic acid (**3**).

Verbascoside (**14**) appears to be a marker for the leaves of the Cima di Melfi variety, while diosmetin glucoside (**20**) instead characterizes the leaves of the Bambina variety.

The above results highlighted the abundance of polyphenolic compounds in olive leaf extracts, in particular oleuropein and its metabolites (hydroxytyrosol and elenolic acid). From a careful study of the LC-MS analyses and from the results reported in the tables, we note the large complexity of the polyphenolic profiles, principally in the Coratina and Leccino varieties. Furthermore, by this approach, it seems that it is not easy to underline differences between secondary metabolites from November crops compared to those from April crops. For this reason, a pseudo-targeted approach was performed by generating a matrix with metabolites identified in the tables by using the method described by Sarais et al. [29] for a more interpretable result and to decrease the number of variables. An analysis of the peak areas of the compounds reported in Table 1 was performed in order to obtain a data matrix based on the negative ionization results.

In each row of the dataset, each marker compound (variable) was represented by an area and each sample (in duplicate) was represented by a column. The resulting score scatter plot and the relative loading plot are reported in Figure 3. Samples collected in November fall in the left area of the plot and leaves collected in April are represented in the right area of the plot, presenting with this approach an evident separation. The loading plot gives us the possibility to recognize, again with different metabolites overlapping in the central part of the plot, that some metabolites distribute in the different areas, and in particular oleuropein (**22**) and citric acid (**2**) are overexpressed in the leaves collected in November, while oleoside (**7**), apigenin glucoside (**19**), and loganic acid (**3**) appear to be overexpressed in samples collected in April. The increase in flavonoid content in olive leaves during spring was attributed to the increase in the biological activity during the leaves’ vegetative cycle renewal, in addition to the need for the plant to increase its chemical defence with the improvement of the phenolic barrier. However, as previously observed by LC-MS analysis, oleuropein could represent a marker of November leaves, and compounds related to its metabolic pathway (oleoside and loganic acid) are markers of leaves collected in April.

We can assume that the biosynthetic pathway is activated in the plant in the spring period, while in November it is present only in the final product of the synthesis, oleuropein.

### 2.3. Phenolic Compound Quantitation

The total phenol content detected by the spectrophotometric assay is reported in Table 2; the interaction of the two variables *C* (cultivar) and *S* (seasonality) significantly influenced the total phenol content. On the whole, in most cases the highest content was detected in November and the cultivars richer in phenols were Bambina and Cima di Mola (1816 and 1788 mg/100 g), followed by Coratina, Leccino, and Cima di Melfi. The collection of olive leaves was carried out in November and April, since these two harvesting times represent two crucial points for the recovery of the biomass according to the olive harvesting for olive oil production (November) and pruning (April).

Different authors have evidenced that the springtime is related to higher contents of phenolic compounds in olive leaves than in the autumn [9,30]. Martinez-Navarro et al. [25] investigated the variation in the main phenolic compound concentration throughout the agronomic cycle, highlighting the monthly variation from month to month. According to our results, in many cases a decrease in phenolic compounds was detected in April.

The main phenolic compounds were quantified by HPLC-DAD and external calibration curves. Oleuropein was the most abundant compound according to different authors, and a higher concentration in November than in April was found [25,30,31]. Similar trends to other studies were also observed during the different seasons, for which during the spring season, the oleuropein content of the considered varieties was lower compared to autumn and winter time [32].

Bambina showed the highest content in oleuropein (1202 mg/100 g), which was was more concentrated in November than in April, followed by Cima di Mola, Coratina, and Leccino, while the lowest content was found in Cima di Melfi. An opposite trend was found for verbascoside, which was more abundant in April than in November, with the highest content in Coratina (142.2 mg/100 g). The different concentration trend of verbascoside was also found by other authors, who highlighted an increase in its concentration in April [7,30].

Luteolin-7-glu content was influenced by season only in Cima di Mola, Cima di Melfi, and Leccino, for which in November the highest concentrations were found; a similar trend was also found for apigenin-7-glu and rutin. According to Lukić et al. [7], luteolin-7-glu was found in the highest concentration among flavonoids.

### 2.4. Antioxidant Activity Evaluation

Several studies have demonstrated the antioxidant activity of olive leaf extracts in different systems from foods to the biopharmaceutical field [33]. The olive leaf extracts were evaluated for their DPPH^•^ and ABTS^•+^ scavenging activities (Figure 4A), and the results of in vitro evaluation assays performed showed that the highest values of antioxidant activity were reached in November, especially in Coratina and Bambina cultivars, followed by Cima di Mola, Cima di Melfi, and Leccino. Lower values in April than in November were found, although the trend among the cultivars was the same. These results were in line with those found for TPC and oleuropein. Significant correlations were found between TPC and the results of antioxidant assays by other authors [34] by highlighting that the antioxidant activity assays were correlated with the content of oleuropein.

The ability of the extracts to improve the stability of purified olive oil was measured by RapidOxy (Figure 4B). Bambina reached the highest oxidative stability of the purified olive oil expressed as induction time with a value of about 80 min, irrespective of the seasonal time. The lowest values were found in Cima di Melfi and Leccino with a value of about 40 min, half of the time reached compared to when the Bambina extract was added in the purified olive oil. According to our previous work, the olive leaf extract is able to carry out an antioxidant activity in lipidic matrices [31]; Orak et al. [34] obtained that olive leaf extracts from different genotypes inhibit the oxidation of β-carotene/linoleic acid emulsion with significant differences among olive leaf genotypes. In this framework, different authors showed the potential of olive leaf extract to be used as in vegetable oils in food matrices as a preservative by inhibiting and/or retarding the oxidation [35].

## 3. Materials and Methods

### 3.1. Raw Materials

Olive leaves were collected from trees (>10 years old) of the cvs. Coratina, Bambina, Leccino, Cima di Melfi, and Cima di Mola grown in the biodiversity repository belonging to the research centre “CRSFA Basile Caramia” and located in the province of Taranto (Apulia, Italy). Samples consisting of mature leaves were collected from the median portions of 1–2 years old branches harvested from the 4 cardinal points of the canopy. Leaves were collected in November 2020 (N) and April (A) 2021, washed with deionized water, and dried in a dryer at 120 °C for 11 min until reaching 2% of moisture content. Dried olive leaves were then pulverized using a grinder. The extraction was performed in duplicate for each harvesting time.

### 3.2. Olive Leaf Extract Preparation

Extracts were obtained using the ultrasound technology. Briefly, 10 g of olive leaves was added to 50 mL of water:ethanol (70:30 *v*/*v*), shaken for 2 min, and sonicated (CEIA, Viciomaggio, Italy) for 20 min at 24 °C. Finally, the suspension was filtered through Whatman (GE Healthcare, Milan, Italy) filter paper and then with nylon filters of 0.45 µm (Sigma Aldrich, St. Louis, MO, USA). The recovered extracts were then subjected to total phenol quantification and antioxidant activity evaluation tests. 

### 3.3. LC-ESI/LTQ-Orbitrap/MS and LC-ESI/LTQ-Orbitrap/MS/MS Analysis

An analytical HPLC method coupled with a hybrid mass spectrometer, which combines the linear trap quadrupole (LTQ) and Orbitrap mass analyser, was used to analyse the secondary polar metabolites of olive leaf extracts. Experiments were achieved using a Thermo Scientific liquid chromatography system constituting a quaternary Accela 600 pump and an Accela autosampler, connected to a Linear Trap-Orbitrap hybrid mass spectrometer (LTQ-Orbitrap XL, Thermo Fisher Scientific, Bremen, Germany) with electrospray ionization (ESI). To separate chromatographically the analytes a Luna C18 5 µm (Phenomenex, Aschaffenburg, Germany 150 × 2 mm) column was used. The mobile phase used was solvent A (water + 0.1% formic acid) and solvent B (acetonitrile + 0.1% formic acid). A linear gradient program at a flow rate of 0.200 mL/min was used: 0–31 min, from 5 to 23% (B); 31–60 min, from 23 to 26% (B); 60–85 min, from 26 to 40% (B); 85–90 min, from 40 to 80% (B); then 5 min to 100% (B) and back to 10% (B) for 5 min. The analyses were performed in negative ion mode and ESI source parameters were as previously described by D’Urso et al. [11].

Xcalibur software version 2.1 was used for instrument control, data acquisition, and data analysis.

### 3.4. Untargeted and Pseudo-Targeted Multivariate Data Analysis

For a comprehensive analysis of the data, a multivariate data analysis was performed by using targeted and untargeted approaches.

For untargeted analysis, the approaches used by D’Urso et al. [11] were applied with slight modifications.

The LC-ESI/LTQOrbitrap/MS chromatograms in negative ion mode were evaluated using the free software package MZmine (http://mzmine.sourceforge.net/ accessed on 21 June 2021), excluding noise from LC-MS profiles. The data with an intensity less than 5.0 × 10^3^ were not considered. Then, a manual peak selection was performed, and 182 peaks were detected. The software generated a data matrix in tabular format (.cvs file). Multivariate data analysis was performed using SIMCA P+ software 12.0 (Umetrix AB, Umea, Sweden) for the principal component analysis (PCA). PCA was performed to define a homogeneous cluster of samples by using the peak area obtained from LC-MS analysis. Pareto scaling was applied to normalize data before multivariate data analysis.

The targeted approach was performed by SIMCA P+ software 12.0 (Umetrix AB, Umea, Sweden) by using PCA, principal component analysis, following the approach used by D’Urso et al. [11]. Pareto scaling was applied before multivariate data analysis.

### 3.5. Total Phenol Content and Antioxidant Activity

Total phenol content was determined according to the Folin–Ciocalteu method. An aliquot of 20 µL of the extract properly diluted and 100 µL of Folin–Ciocalteu reagent were added to 980 µL of H_2_O Milli-Q^®^. After an incubation of 3 min, 800 µL of 7.5% of Na_2_CO_3_ was added, and the solution was incubated in the dark for 1 h. The absorbance was measured at 720 nm and the results were expressed as mg of gallic acid equivalent per 100 g of dried olive leaves. The analyses were carried out in triplicate.

Regarding antioxidant tests, DPPH and ABTS assays were carried out, which evaluated the capacity of the extract to scavenge the stable DPPH radical and to inhibit the ABTS radical (ABTS^+^) compared with a reference antioxidant standard (Trolox), respectively. The DPPH assay was conducted adding 50 µL of the extract, properly diluted, to 950 µL of the DPPH solution. When the incubation time elapsed after 30 min in the dark, the absorbance was measured at 517 nm using a Cary 60 Agilent spectrophotometer (Cernusco, Milan, Italy). ABTS assay included the addition of 50 µL of the extract to 950 µL of ABTS solution. After 8 min in the dark, the absorbance was measured at 734 nm and the results expressed as mmol TE per 100 g of dried olive leaves. The analyses were carried out in triplicate.

### 3.6. HPLC-DAD Analysis

The phenolic profiles of olive leaf extracts were characterized by high-performance resolution liquid chromatography coupled to a diode-array-detector (DAD) according to Difonzo et al. [31]. The analysis was performed using the UHPLC Dionex Ultimate 3000 system equipped with an LPG-3400 RS quaternary pump, WPS-3000 TRS autosampler, vacuum degasser, DAD, and TCC-3000 RS column oven. The column used was RP-C18 AcclaimTM 120-Thermo Fisher, 150 × 3 mm length, particle size 3 µm, maintained at a temperature of 35 °C. The mobile phase consisted of (A) water/acetic acid (98:2 *v*/*v*) and (B) acetonitrile. The flow rate was 1 mL/min, and the gradient program of solvent B was as follows: 0–5 min 5% solvent B, 5–10 min 20% solvent B, 10–15 min 25% solvent B, 15–20 min 35% solvent B, 20–25 min 100% solvent B, and 25–35 min 5% solvent B. The polyphenol quantification was performed using calibration curves of external standard, considering a wavelength range of 240–350 nm. The linear range for oleuropein (500–5000 mg/L), verbascoside (100–3000 mg/L), rutin, apigenin-7-glucoside, and luteolin-7-glucoside (35–1000 mg/L) concentrations were evaluated. In linearity ranges were used 10 different standard solutions of the lower and upper limit concentration. Calibration curves were obtained by plotting the area of external standard against the known concentration of each compound, and each concentration of standard solutions was analysed in triplicate. A good linearity with correlation coefficients (*R*^2^) in the range of 0.9983 to 0.9999 was achieved for all analytes.

The limit of detection (LOD) and the limit of quantification (LOQ) for each target standard compound were determined, under the optimized conditions, by linear regression considering LOD = 3S_a_/*b* and LOQ = 10S_a_/*b*, where S_a_ is the standard deviation of the response and *b* is the slope of the calibration curve. The LOD and LOQ calculated for oleuropein were 69 and 212 mg/L, for verbascoside 40 and 82 mg/L; for rutin, apigenin-7-glucoside, and luteolin-7-glucoside LOD varied from 7–15 mg/L and LOQ from 18–29 mg/L.

The results were expressed as mg per 100 g of dried olive leaves.

### 3.7. Oxidative Stability Test

RapidOxy (Anton Paar, Blankenfelde-Mahlow, Germany) test was used for the measurement of the oxidative stability. RapidOxy is a microprocessor-controlled automatic testing device for quick measures of the oxidative stability of lipid matrices in response to forced oxidation with the increase in temperature and O_2_ pressure. The induction time of the sample is measured as the time needed for a 10% drop in the oxygen pressure. The set parameters were the following: T = 140 °C and P = 700 kPa. An aliquot of 100 µL of the obtained extracts were added to purified olive oil and emulsified. Each sample was analysed in triplicate.

## 4. Conclusions

Our results highlighted the abundance of polyphenolic compounds in olive leaves, in particular oleuropein and its metabolites (hydroxytyrosol and elenolic acid). Large complexity in the polyphenolic profile was found mainly in the Coratina and Leccino varieties. By using a pseudo-targeted approach, it was found that oleuropein and citric acid are overexpressed in the leaves collected in November, while oleoside, apigenin glucoside, and loganic acid appear to be overexpressed in samples collected in April. These results were similar to those reported by other authors, where the increase in flavonoid content in olive leaves during spring was attributed to the increase in the biological activity during the leaves’ vegetative cycle renewal [36,37]. Most likely, the biosynthetic pathway is activated for the plant in the spring period, while in November it is possible to define only the presence as a biological marker of the final product of the synthesis, oleuropein.

The results of the quantitation of the main phenolic compounds detected highlighted the highest level of oleuropein in Bambina, followed by Cima di Mola, Coratina, and Leccino, with the highest concentration in November. Moreover, the results of the total phenol content quantitation showed the highest content in November, and the cultivars richer in phenols were Bambina and Cima di Mola, followed by Coratina, Leccino, and Cima di Melfi; a similar trend was found for the antioxidant activity evaluation and oxidative stability in the lipidic fraction. Overall, our study allowed us to identify the metabolomic profiles of the leaf extracts from five olive cultivars grown in the Apulia region in two different seasonal times, giving valuable information to better upcycle this by-product for the production of added-value formulations in food and pharmaceutical sectors.

This approach could contribute to make the olive oil chain more sustainable.

## Figures and Tables

**Figure 1 plants-11-03321-f001:**
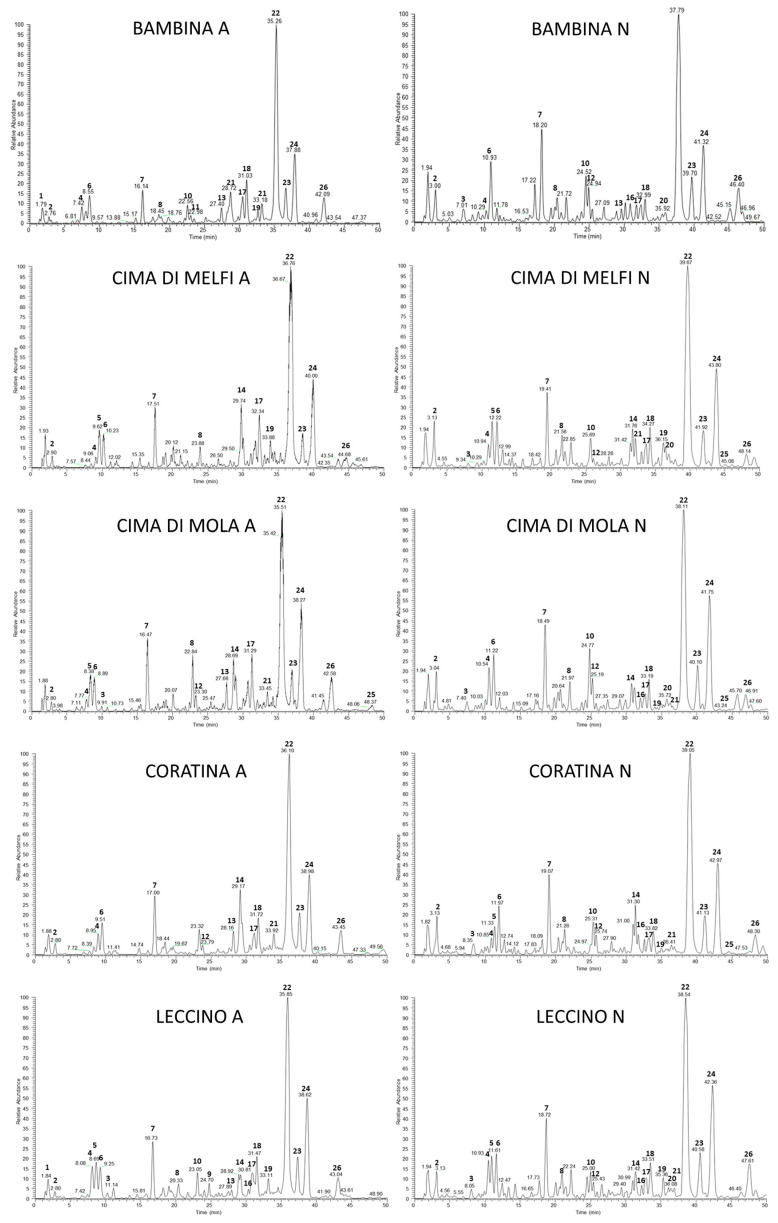
LC-ESI/LTQ-Orbitrap/MS profiles of *Olea europea* hydro-alcoholic extracts from the November (N) and April (A) leaves of Bambina, Cima di Melfi, Cima di Mola, Coratina, and Leccino cultivars (negative mode).

**Figure 2 plants-11-03321-f002:**
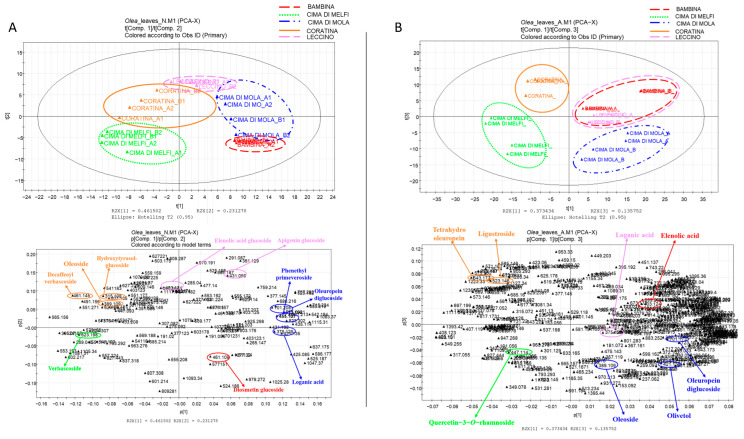
Principal component analysis (PCA) of hydro-alcoholic extracts of *Olea europea* April (**A**) and November (**B**) leaf extracts. Score scatter plot at the top and loading scatter plot at bottom. In loading scatter plot, biomarkers of any cultivar are underlined.

**Figure 3 plants-11-03321-f003:**
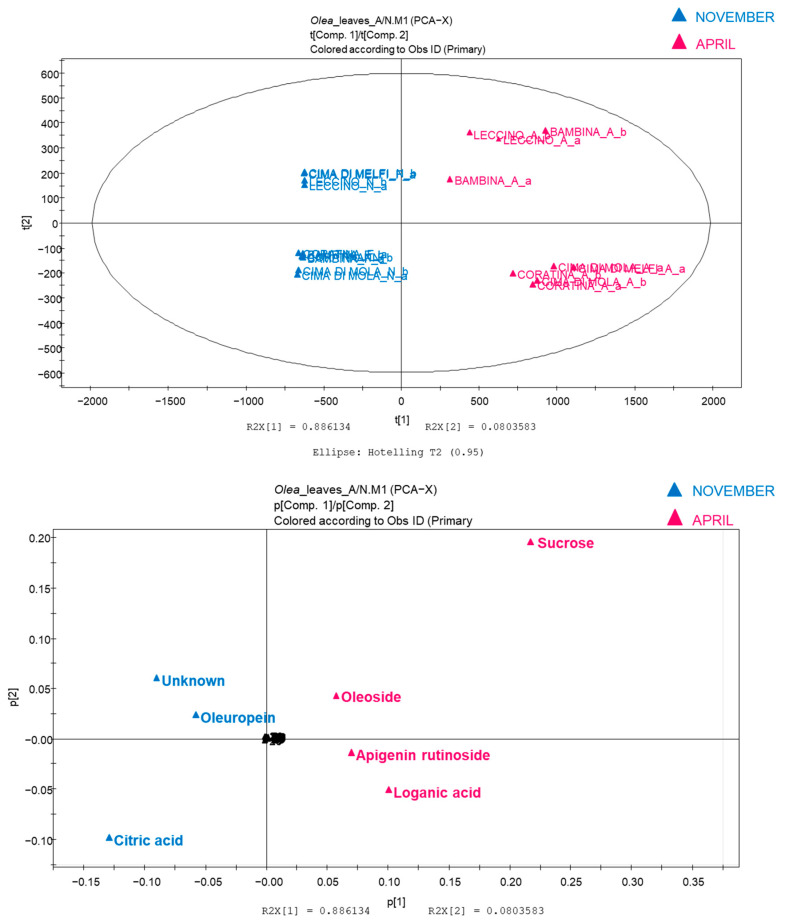
Principal component analysis (PCA) of hydro-alcoholic extracts of olive leaves (pseudo-targeted approach). Score scatter plot and loading scatter plot are reported: A, April; N, November.

**Figure 4 plants-11-03321-f004:**
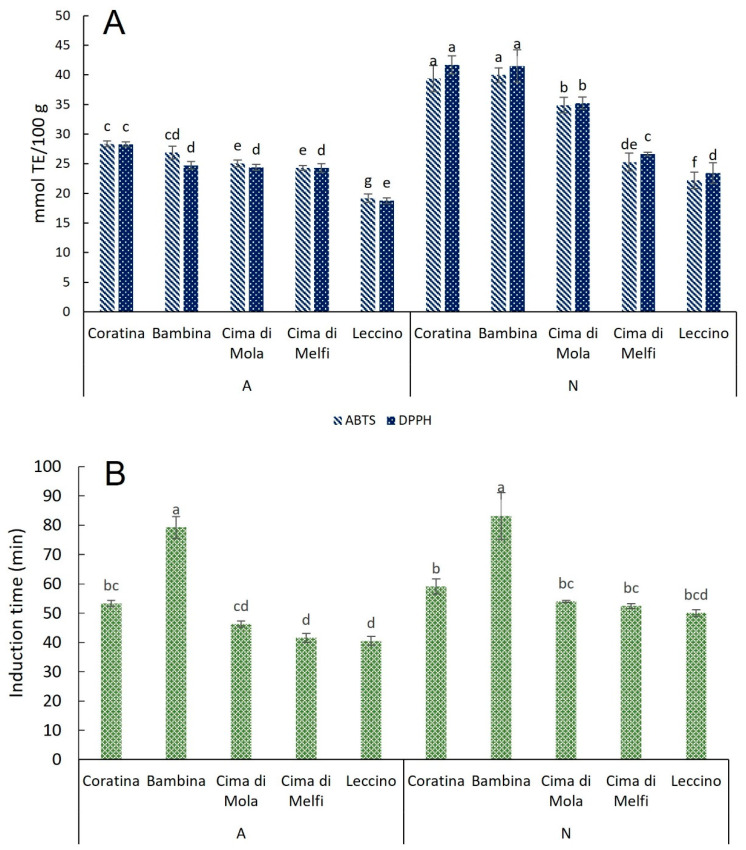
Mean values, standard deviations, and results of statistical analysis (two-way ANOVA) of the antioxidant activity evaluation on olive leaf extracts (**A**) and RapidOxy evaluation (**B**). Different letters indicate significant differences at *p* < 0.05.

**Table 2 plants-11-03321-t002:** Total phenol content determination and main phenolic compounds’ quantitation by HPLC-DAD.

	Coratina		Bambina		Cima Di Mola		Cima Di Melfi		Leccino		*p*-Value
Parameters	A	N	A	N	A	N	A	N	A	N	C*S
TPC	1733.0±19.1 ^b^	1742.6 ±18.7 ^b^	1349.1 ±9.5 ^d^	1816.7 ±10.8 ^a^	1262.6 ±5.6 ^e^	1788.5 ±27.4 ^a^	1241.4 ±14.2 ^e^	1542.3 ±14.4 ^c^	928.4 ±12.6 ^f^	1691.0 ±8.6 ^b^	*p* < 0.001
Rutin	21.3 ±0.1 ^b^	24.9 ±0.2 ^a^	14.0 ±0.1 ^d^	20.7 ±0.3 ^c^	9.2 ±0.3 ^g^	12.9 ±0.2 ^e^	4.6±0.1 ^j^	6.6 ±0.1 ^h^	5.3 ±0.1 ^i^	10.9 ±0.2 ^f^	*p* < 0.001
Verbascoside	142.2 ±1.3 ^a^	95.2 ±0.3 ^c^	90.5 ±1.1 ^cd^	68.6 ±2.2 ^e^	92.2 ±2.3 ^cd^	69.6 ±0.3 ^e^	108.0 ±2.0 ^b^	89.7 ±1.7 ^cd^	87.7 ±2.8 ^d^	59.3 ±1.3 ^f^	*p* < 0.001
Luteolin-7-glu	31.4 ±0.2 ^b^	29.9 ±0.1 ^bc^	27.5 ±0.3 ^c^	28.8 ±0.5 ^c^	13.7 ±0.4 ^e^	17.8 ±0.2 ^d^	32.2 ±1.3 ^b^	35.8 ±0.3 ^a^	12.4 ±1.1 ^e^	20.1 ±0.2 ^d^	*p* < 0.001
Apigenin-7-glu	11.3 ±0.1 ^ab^	11.4 ±0.1 ^ab^	11.9 ±0.3 ^ab^	12.7 ±0.6 ^ab^	7.8 ±0.4 ^cd^	13.0 ±0.3 ^a^	8.9 ±1.1 ^c^	10.9 ±0.2 ^b^	2.4 ±0.5 ^e^	6.1 ±0.1 ^d^	*p* < 0.001
Oleuropein	864.4 ±1.70 ^e^	1037.8 ±55.9 ^bc^	878.8 ±17.7 ^de^	1202.0 ±12.4 ^a^	764.9 ±13.6 ^f^	1078.8 ±2.1 ^b^	686.8 ±13.2 ^fg^	905.2 ±3.6 ^de^	620.4 ±16.8 ^g^	956.8 ±12.0 ^cd^	*p* < 0.01
Oleuropein isomers	96.5 ±0.9 ^c^	130.2 ±1.5 ^b^	78.2 ±2.2 ^cd^	137.5 ±5.9 ^b^	85.4 ±3.4 ^c^	156.3 ±0.5 ^a^	75.7 ±0.5 ^cd^	129.5 ±0.8 ^b^	67.5 ±0.1 ^d^	160.9 ±8.0 ^a^	*p* < 0.001

The mean values are expressed in mg/100 g of dried leaves. Two-way ANOVA and Tukey’s Test were performed to compare the samples by considering the variables C (cultivar) and season (S). Different letters (^a–g^) in the same row mean significant differences at *p* < 0.05. Abbreviations: A, April; N, November; TPC, total phenol content.

## Data Availability

The data are available on request.

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
