# Peer review of "Metabolomics Approach to Characterize Green Olive Leaf Extracts Classified Based on Variety and Season"

_plants, 2022, doi:10.3390/plants11233321_

Round 1

Reviewer 1 Report

The manuscript presents interesting and valuable work within the scope of the Plants. Lots of work has been done. From a scientific point of view, the manuscript is well-organized and needs no substantial changes. However, several considerations should be taken into account before publication. Below you can find a list of my comments and suggestions:

Abstract part is quite general. Please, correct it.

3. Materials and Methods

3.6. HPLC analysis

The authors should add a reference for the HPLC method because this method is not new.

Did the authors previously validate the analytical methods applied for polyphenols?

The authors must include the validation parameters: range of linearity, correlation coefficient, accuracy, repeatability, precision, limits of quantification and detection, selectivity, and specificity.

The authors should explain how the accuracy of the applied method was checked.

Figure 2. needs to be clarified. Please, correct it.

Unfortunately, many mistakes affect the reading and understanding of this manuscript. Several typing errors should be corrected (line 308: “0.1 %”; line 309: “0.200 mL/min”)

The proposed manuscript is acceptable for publication by Plants after minor revisions.

Author Response

Dear reviewer, 

please find attached our revisions.

Best regards

Reviewer 2 Report

The article under appreciation is an interesting contribution and the study is well performed but some improvements could be made:

Abstract-Please include some results obtained for total phenol content and antioxidant activity and a phrase related to the statistical treatment of the results

Introduction- The authors can underline the current state of research on this topic presented in the literature. Authors are asked to highlight the originality of the work and show what they bring new with their experimental determinations that they have achieved.

The authors used the words: polyphenols and biophenols. An explanation regarding these two words must be inserted.

Method- The method used must be validated, in this case data about precision, robustness, recovery, working range, LOD, LOQ should be added.

Author Response

(The authors gave the same response as above.)

Reviewer 3 Report

The manuscript is well structured, and the methodology is clear, however the novelty is poor. I think that may be is appropriate for a technical note. In my opinion the authors have an excellent mass spectrometer but, may be, information more deep could be obtained for explain metabolic process.  Then, the general considerations are detailed above:  

General consideration for authors:

When I read that the sample leaves were pick up in two different dates, I hope to find an explanation about this decision and a deep discuss about the metabolic aspects. However, there are not these explanations and only an analytical discussion is observed. In my opinion is totally necessary introduce this kind of topics, principally considering the journal scope.

I recommend avoid the paragraph with only one sentences.

Sentences like as "It could therefore be more advantageous to use the by-product coming from these two varieties, rather than the one obtained Cima di Mola, Cima di Melfi or Bambina varieties." need other types study for its formulation. In my opinion the authors doing some affirmations without clear evidence.

The concentration should be expressed according to the samples, is say in mass units, because in this way we can compared with other works and is representative to sample, that are leaves and not a liquid.

The references are too much.

Author Response

(The authors gave the same response as above.)

Round 2

Reviewer 3 Report

Dear Authors,

I am reading the responses, however I found few changes doing on manuscript. I understand that the authors considerer to make a metabolite networking for future work, but it is important improve the actual manuscript, to be successful publication in the present Journal.

In this way, I recommend that it is necessary improve the discussion about biological aspect, the authors added one sentences in the introduction section, but it is necessary improve the discussion. Newly, the authors response that “The metabolic aspects should be treated in a next work.”. Then, when this response is repeat in each request when the reviewer asked about more information, I think strongly, that the present is an “incomplete work”. For this reason, it is mandatory that more details to be included in the present manuscript.

Respect to concentration, expressed in mg/L, the authors will understand that if they present this work, as an aim to valorise olive leaves, it is important express the bioactive compounds according to these, because the leaves are the subject and not the hydroalcoholic extract.  

Then, it is necessary to continuo with the improve of the present manuscript to be considered to its publication in the present journal.

Author Response

Dear reviewer,

please find attached our responses.

Best regards

Round 3

Reviewer 3 Report

The authors have included the requested points, for this reason, now it is possible its publication.